# State Observer Based on an Accelerometer for an Elastic Joint with Nonlinear Friction

Kwang-Hee Lee [1,2], Hyungjong Kim [2,*] and Tae-Yong Kuc [1]

1 Department of Electrical and Computer Engineering, Sungkyunkwan University, Suwon 16419, Republic of Korea
2 Korean Institute of Industrial Technology, Ansan 15588, Republic of Korea
* Correspondence: hyungjong@gmail.com; Tel.: +82-31-8040-6384; Fax: +82-31-8040-6390

**Abstract:** This paper presents a state observer for an elastic joint with nonlinear friction via the information from an acceleration sensor. In order to avoid discontinuities, the nonlinear friction of the motor, which includes static, coulomb, and viscous terms, is considered a smooth function. In addition, it uses an acceleration sensor to obtain the information about the link with high uncertainty. The proposed state observer guarantees that the estimation error for the position and velocity of the link connected via an elastic joint containing a nonlinear stiffness (elasticity) converges to zero. In addition, it is shown that the observer gain can be designed by LMI (linear matrix inequality) optimization. Finally, to verify the performance of the proposed observer, the method proposed in this paper is tested by experiments on a two-inertia system with an elastic shaft.

**Keywords:** elastic joint; state observer; acceleration sensor; nonlinear friction; Lipschitz; linear matrix inequality (LMI)

## 1. Introduction

The design of a robust controller for elastic joint manipulators has attracted much attention over the past several decades. The importance of the control applied to a system is increased when precise and rapid movement needs to be obtained. To obtain high control performance, the precise positioning problem of the elastic joint has been studied for a long time. Several effective control strategies have been devised in [1–4]. The majority of previously proposed methods need information on the exact state of the joint (position, speed, etc.). However, the research on state estimation is extremely low. In particular, due to the nonlinear characteristics of elastic joints, it is not easy to find the states (e.g., position or velocity of link) of the link side under the constraint that only the states of the motor are known [5,6].

In recent years, there have been several attempts to solve the problem of estimating the position and velocity in mechanical systems [7–9]. In particular, the authors in [9] present a robust state observer for elastic joints that contain nonlinear functional components in the stiffness part. The state observer was designed using the results of [10–12]. However, this study does not consider the effect of frictional forces. As depicted in Figure 1, elastic joints generally have a nonlinear friction torque that is approximated as affecting the motor side only. The nonlinear characteristics of its friction appear to be especially striking in the low-speed region of the motor [13].

In this brief, we consider the nonlinear frictional force which was not considered in [9]. We present a state observer that overcomes the problem of the state estimation error not being eliminated due to the nonlinear friction. In general, the nonlinear friction model is discontinuous because of the signum function in coulomb friction [14,15]. In order to avoid the discontinuities, the nonlinear friction of the motor, which includes the static, coulomb, and viscous terms, is considered a smooth function. The proposed state observer guarantees that the estimation error for the position and velocity of the link connected via an elastic

joint containing a nonlinear stiffness (elasticity) converges to zero. In addition, it is shown that the observer gain can be designed by LMI (linear matrix inequality) optimization. Finally, we conduct experiments on a two-inertia system consisting of two motors.

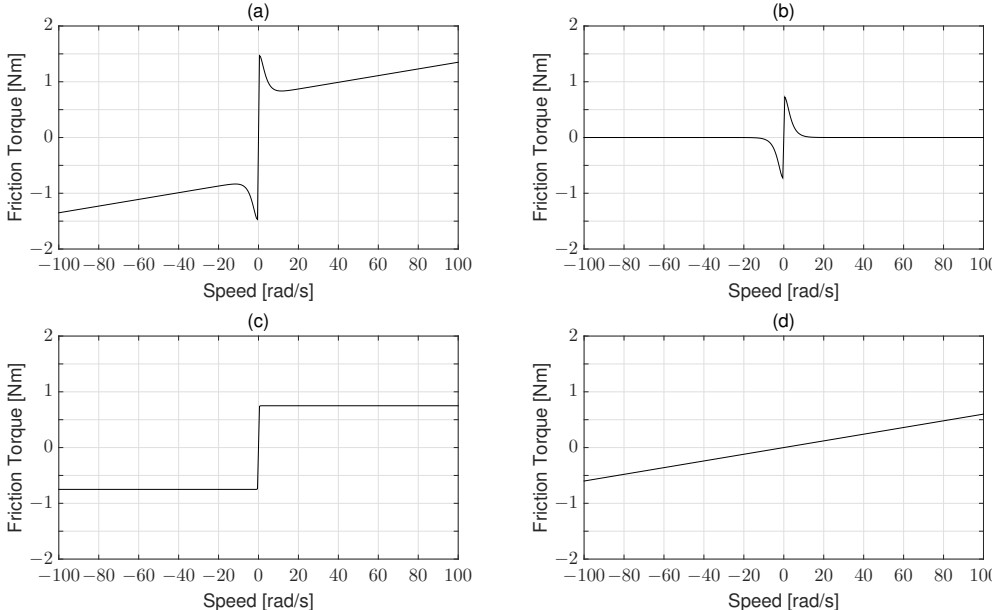

**Figure 1.** Example of nonlinear friction: (**a**) static + coulomb + viscous friction, (**b**) static friction, (**c**) coulomb friction, (**d**) viscous friction.

The rest of this paper is organized as follows: We introduce the problem formulation and provide a design method of the state observer in Section 2. Next, Section 3 shows the stability and performance analysis of the proposed observer. In Section 4, we experimentally test the proposed observer on a servo drive system.

## 2. Problem Formulation and Solution

We consider an elastic joint model with nonlinear friction and stiffness, as described by [7]:

$$\begin{aligned}
J_l(\theta_l)\ddot{\theta}_l + D(\dot{\theta}_l - \dot{\theta}_m) - K(\theta_l, \theta_m) + C(\theta_l, \dot{\theta}_l) + G(\theta_l) &= 0, \\
J_m\ddot{\theta}_m + D(\dot{\theta}_m - \dot{\theta}_l) + K(\theta_l, \theta_m) + F_m(\dot{\theta}_m) &= \tau.
\end{aligned} \tag{1}$$

where $\theta_l$ and $\theta_m$ are the angular positions of the link-side and motor-side, respectively. $J_l(\theta_l)$ and $G(\theta_l)$, which depend on the position of the link, are the link inertia and the gravity term, respectively. While the coriolis and centrifugal term $C(\theta_l, \omega_l)$ depend on the position and velocity of the link, the motor inertia $J_m$ and the damping $D$ have a constant value. $\tau$ is the torque input to the motor. As shown in Figure 1, the nonlinear friction on the motor side $F_m(\omega_m)$ is given by:

$$F_m(\omega_m) = f_v \omega_m + f_c(\mu_k + (1 - \mu_k)\operatorname{sech}(\beta\omega_m))\tanh(\alpha\omega_m) \tag{2}$$

where $f_v$ and $f_c$ are the kinetic coefficients of viscous and coulomb friction, respectively. $\alpha$, $\beta$ and $0 < \mu_k < 1$ are the suitable parameters of the nonlinear friction. Moreover, as shown in [7,9], the stiffness $K(\theta_l, \theta_m)$ with nonlinear characteristics is expressed by

$$K(\theta_l, \theta_m) = \begin{cases}
-k_1\theta_B - k_2\theta_B^3 - (k_1 + 3k_2\theta_B^2)(-\theta_m + \theta_l - \theta_B), & \text{if } \theta_m - \theta_l < -\theta_B. \\
k_1(\theta_m - \theta_l) + k_2(\theta_m - \theta_l)^3, & \text{if } \|\theta_m - \theta_l\| \leq \theta_B. \\
k_1\theta_B + k_2\theta_B^3 + (k_1 + 3k_2\theta_B^2)(\theta_m - \theta_l - \theta_B), & \text{if } \theta_m - \theta_l > \theta_B.
\end{cases} \tag{3}$$

where $k_1$, $k_2$, and $\theta_B$ are all position numbers. Here, $k_1$ and $k_2$ mean linear and nonlinear stiffness coefficients, respectively. $\theta_B$ is a break-point, which means the physical limit by the torsion degree between the motor and the link. The representation of (1) can be obtained as follows:

$$
\begin{aligned}
\dot{\theta}_l &= \omega_l, \\
\dot{\omega}_l &= J_l^{-1}(\theta_l)(D(\omega_m - \omega_l) + K(\theta_l, \theta_m) - C(\theta_l, \omega_l) - G(\theta_l)), \\
\dot{\theta}_m &= \omega_m, \\
\dot{\omega}_m &= J_m^{-1}(D(\omega_l - \omega_m) - K(\theta_l, \theta_m) - F_m(\omega_m)) + J_m^{-1}\tau.
\end{aligned}
\tag{4}
$$

Here, $\omega_l$ and $\omega_m$ are the derivatives of $\theta_l$ and $\theta_m$, which represent the angular velocities of the link and motor, respectively.

The final goal of this study is to design a robust state observer that guarantees the asymptotic estimation performance for all the states of the Equation (4). Now, we pose some conditions in order to solve the problem.

**Assumption 1.** *The position of motor-side $\theta_m$ and the acceleration of link-side $\dot{\omega}_l$ are measurable while the link position $\theta_l$, the link velocity $\omega_l$, and the motor velocity $\omega_m$ are not.*

**Assumption 2.** *The motor inertia $J_m$ and the damping parameter $D$ and all parameters of the nonlinear friction (2) and stiffness (3) are known.*

Now, let

$$
x := \begin{bmatrix} x_1 & x_2 & x_3 & x_4 \end{bmatrix}^\top := \begin{bmatrix} \theta_l & \omega_l & \theta_m & \omega_m \end{bmatrix}^\top.
$$

Then, the observer design problem considered in this paper can be stated as follows. For the given system (1), find a state observer given in the form

$$
\dot{\hat{x}}(t) = f(t, \hat{x}, \theta_m, \dot{\omega}_l, \tau)
\tag{5}
$$

such that $\hat{x}(t)$ converges to $x(t)$ as time goes to infinity.

Inspired by [12], a solution to the problem proposed in the above is given in terms of a state observer for a Lipschitz nonlinear system, which will be described below. First of all, it follows from (4) with (2) and (3) that

$$
\begin{aligned}
\dot{x} &= Ax + Y + \Omega(x) + Bu \\
y_1 &= Cx
\end{aligned}
\tag{6}
$$

where $u := \tau$ is the input torque and $y := \begin{bmatrix} y_1 & y_2 \end{bmatrix}^T := \begin{bmatrix} \theta_m & \dot{\omega}_l \end{bmatrix}^T$ are the values measurable by Assumption 1, and

$$
A = \begin{bmatrix}
0 & 1 & 0 & 0 \\
0 & 0 & 0 & 0 \\
0 & 0 & 0 & 1 \\
J_m^{-1}k_1 & J_m^{-1}D & -J_m^{-1}k_1 & -J_m^{-1}(D+f_v)
\end{bmatrix}, \quad
Y = \begin{bmatrix} 0 \\ y_2 \\ 0 \\ 0 \end{bmatrix},
$$

$$
\Omega(x) = \begin{bmatrix} 0 \\ 0 \\ 0 \\ -J_m^{-1}(\phi(x_1, x_3) + \psi(x_4)) \end{bmatrix}, \quad
B = \begin{bmatrix} 0 \\ 0 \\ 0 \\ J_m^{-1} \end{bmatrix}, \quad
C = \begin{bmatrix} 0 \\ 0 \\ 1 \\ 0 \end{bmatrix}^T,
$$

$$
\phi(x_1, x_3) = \begin{cases}
\phi_1(x_1, x_3) = 3k_2\theta_B^2(x_3 - x_1) + 2k_2\theta_B^3, & \text{if } x_3 - x_1 < -\theta_B, \\
\phi_2(x_1, x_3) = k_2(x_3 - x_1)^3, & \text{if } \|x_3 - x_1\| \le \theta_B, \\
\phi_3(x_1, x_3) = 3k_2\theta_B^2(x_3 - x_1) - 2k_2\theta_B^3, & \text{if } x_3 - x_1 > \theta_B,
\end{cases}
$$

$$
\psi(x_4) = f_v x_4 + f_c(\mu_k + (1 - \mu_k)\text{sech}(\beta x_4))\tanh(\alpha x_4)
$$

Next, we present a state observer for the system (6) as follows:

$$\dot{\hat{x}} = A\hat{x} + Y + \Omega(\hat{x}) + Bu + L(y_1 - \hat{y}_1)$$
$$\hat{y}_1 = C\hat{x} \tag{7}$$

where the observer gain $L$ is

$$L = \frac{P^{-1}C^\top}{2\epsilon}. \tag{8}$$

Here, the matrix $P > 0$ is chosen such that for some small $\epsilon > 0$, the following LMI (linear matrix inequality) [16] holds:

$$\begin{bmatrix} A^\top P + PA + \gamma^2 I - \frac{1}{\epsilon}C^\top C & P \\ P & -I \end{bmatrix} < 0. \tag{9}$$

Now, we state the main result of this note, the proof of which is given in the following section, along with a detailed explanation about the proposed observer.

**Theorem 1.** *Under Assumption 1 and 2, the state observer (7) guarantees that the estimation error $e(t) := x(t) - \hat{x}(t)$ converges to zero as time goes to infinity.*

### 3. Stability and Performance Analysis

In order to prove Theorem 1, we obtain that the nonlinear function $\Omega(x)$ of (6) has the following property.

**Lemma 1.** *$\Omega(x)$ of (6) is globally Lipschitz. In other words, there exists a positive number $\gamma$ (the so-called Lipschitz constant) that satisfies the following Lipschitz condition:*

$$\|\Omega(x) - \Omega(\hat{x})\| \le \gamma \|x - \hat{x}\|, \quad {}^\forall x, \hat{x} \in \mathbb{R}^4.$$

*In addition, the Lipschitz constant is expressed as:*

$$\gamma = J_m^{-1}(6k_2\theta_B^2 + \alpha f_c). \tag{10}$$

**Proof.** The Jacobian matrix of $\Omega$ at $x$ is given by:

$$\frac{\partial \Omega}{\partial x} = \begin{bmatrix} 0 & 0 & 0 & 0 \\ 0 & 0 & 0 & 0 \\ 0 & 0 & 0 & 0 \\ -J_m^{-1}\frac{\partial\phi(x_1,x_3)}{\partial x_1} & 0 & -J_m^{-1}\frac{\partial\phi(x_1,x_3)}{\partial x_3} & -J_m^{-1}\frac{\partial\psi(x_4)}{\partial x_4} \end{bmatrix}$$

where

$$\frac{\phi(x_1,x_3)}{\partial x_1} = \begin{cases} \frac{\phi_1(x_1,x_3)}{\partial x_1} = -3k_2\theta_B^2, & \text{if } x_3 - x_1 < -\theta_B, \\ \frac{\phi_2(x_1,x_3)}{\partial x_1} = -3k_2(x_3 - x_1)^2, & \text{if } \|x_3 - x_1\| \le \theta_B, \\ \frac{\phi_3(x_1,x_3)}{\partial x_1} = -3k_2\theta_B^2, & \text{if } x_3 - x_1 > \theta_B, \end{cases}$$

$$\frac{\phi(x_1,x_3)}{\partial x_3} = \begin{cases} \frac{\phi_1(x_1,x_3)}{\partial x_3} = 3k_2\theta_B^2, & \text{if } x_3 - x_1 < -\theta_B, \\ \frac{\phi_2(x_1,x_3)}{\partial x_3} = 3k_2(x_3 - x_1)^2, & \text{if } \|x_3 - x_1\| \le \theta_B, \\ \frac{\phi_3(x_1,x_3)}{\partial x_3} = 3k_2\theta_B^2, & \text{if } x_3 - x_1 > \theta_B, \end{cases}$$

$$\frac{\partial\psi(x_4)}{\partial x_4} = \alpha f_c\mu_k\text{sech}^2(\alpha x_4)$$
$$+ f_c(1 - u_k)\left(-\beta\text{sech}(\beta x_4)\tanh(\beta x_4)\tanh(\alpha x_4) + \alpha\text{sech}(\beta x_4)\text{sech}^2(\alpha x_4)\right)$$
$$\le \alpha f_c\mu_k + f_c(1 - \mu_k)\alpha = \alpha f_c.$$

Thus, the function $\Omega(\cdot)$ is continuously differentiable on $\mathbb{R}^4$. In addition, the $\frac{\partial \Omega}{\partial x}$ is uniformly bounded as follows:

$$\left\| \frac{\partial \Omega}{\partial x} \right\|_\infty = J_m^{-1} \left( \left| \frac{\phi(x_1, x_3)}{\partial x_1} \right| + \left| \frac{\phi(x_1, x_3)}{\partial x_3} \right| + \left| \frac{\partial \psi(x_4)}{\partial x_4} \right| \right) \leq J_m^{-1} \left( 6k_2 \theta_B^2 + \alpha f_c \right).$$

Therefore, the proof is complete from ([17] Lemma 3.3). □

Now the dynamics for the estimation error $e(t)$ can be expressed as follows:

$$\begin{aligned} \dot{e} &= \dot{x} - \dot{\hat{x}} \\ &= (A - LC)e + (\Omega(x) - \Omega(\hat{x})). \end{aligned} \tag{11}$$

Consider a quadratic Lyapunov function candidate

$$V(t) = e^\top(t) P_1 e(t).$$

Then, it follows from Lemma 1 that the derivative of $V(t)$ is given by:

$$\begin{aligned} \dot{V} &= e^T((A - LC)^\top P_1 + P_1(A - LC))e + 2e^T P_1(\Omega(x) - \Omega(\hat{x})) \\ &\leq e^T((A - LC)^\top P_1 + P_1(A - LC))e + 2\|P_1 e\| \|(\Omega(x) - \Omega(\hat{x}))\| \\ &\leq e^T((A - LC)^\top P_1 + P_1(A - LC))e + 2\gamma \|P_1 e\| \|e\| \\ &\leq e^T((A - LC)^\top P_1 + P_1(A - LC))e + \gamma^2 e^T P_1 P_1 e + e^T e \\ &= e^T((A - LC)^\top P_1 + P_1(A - LC) + \gamma^2 P_1 P_1 + I)e. \end{aligned} \tag{12}$$

Next, let $P_1 = \gamma^{-1} P$; it then follows from the Schur complement that the LMI (9) can be written as:

$$\begin{aligned} &A^\top P + PA + \gamma^2 I - \frac{1}{\epsilon} C^\top C + PP < 0 \\ &\Leftrightarrow (A - LC)^\top P_1 + P_1(A - LC) + \gamma^2 P_1 P_1 + I < 0 \end{aligned} \tag{13}$$

where $L = \frac{P^{-1} C^\top}{2\epsilon}$. From Equations (12) and (13), $\dot{V} < 0$, and the error dynamics (11) are asymptotically stable by ([17] Theorem 4.1), i.e., $\lim_{t \to \infty} e(t) = 0$.

In actual system, the acceleration $y_2$ of the vector $Y$ contains measurement noise. On the other hand, the encoder noise for motion position $y_1$ is small enough to be ignored. For the optimal design for the measurement noise, instead of (6), we consider the noise corrupted system:

$$\begin{aligned} \dot{x} &= Ax + Y + \Omega(x) + Bu + Y_d d(t) \\ y_1 &= Cx \end{aligned} \tag{14}$$

where $d(t) \in \mathbb{R}$ is measurement noise for acceleration and $Y_d := \begin{bmatrix} 0 & 1 & 0 & 0 \end{bmatrix}^T$. Now, we consider the minimization of the induced $L_2$ gain between the acceleration measurement noise $d$ and the estimation error $e$, i.e., $\|H_{d \to e}\|_\infty$.

**Theorem 2.** *For the given system (14) and the proposed observer (7), it is supposed that the noise $d(t)$ is bounded. Then, it follows from the observer gain $L = \frac{P^{-1} C^\top}{2\epsilon}$ that $\|H_{d \to e}\|_\infty \leq \kappa$ if $P > 0, \epsilon > 0$, and $\kappa \geq 0$ such that*

$$\begin{bmatrix} A^\top P + PA + (1 + \gamma^2)I - \frac{1}{\epsilon} C^\top C & P & PY_d \\ P & -I & 0 \\ Y_d^T P & 0 & -\kappa^2 I \end{bmatrix} < 0. \tag{15}$$

The detailed proof is omitted since it is similar to the proof of ([12] Theorem 5). From the above result, it is noted that the observer gain $L$ can be designed by the solution $P$ of the LMI (15) for a given $\epsilon > 0$ and $\kappa \geq 0$.

## 4. Experiment Results

Even though our goal is to observe all states of the elastic joint expressed as (1), we carry out experiments for a two-inertia system shown in Figure 2 to verify the effectiveness of the proposed observer in the previous section because it represents quite well the dynamic properties of the manipulator having a flexible joint [18]. As shown in Figure 3, a two-motor system consisting of a driving motor and a load motor is illustrated in this experiment. The two motors connected with an elastic shaft and a friction adjustment on the driving motor side are similar to [19]. The proposed observer has been implemented in a high-performance embedded motion controller (manufactured by National Instruments Corporation) with servo motor drivers (manufactured by Panasonic Industry Corporation).

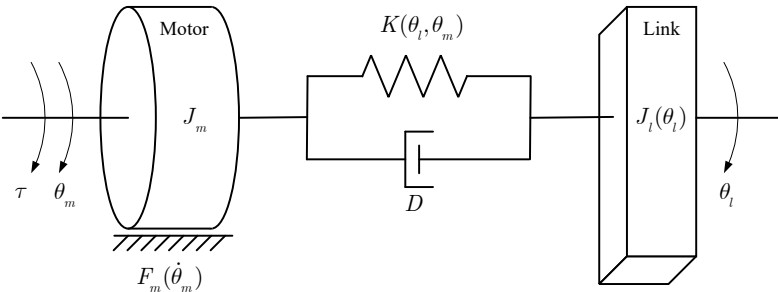

**Figure 2.** The two-inertia system.

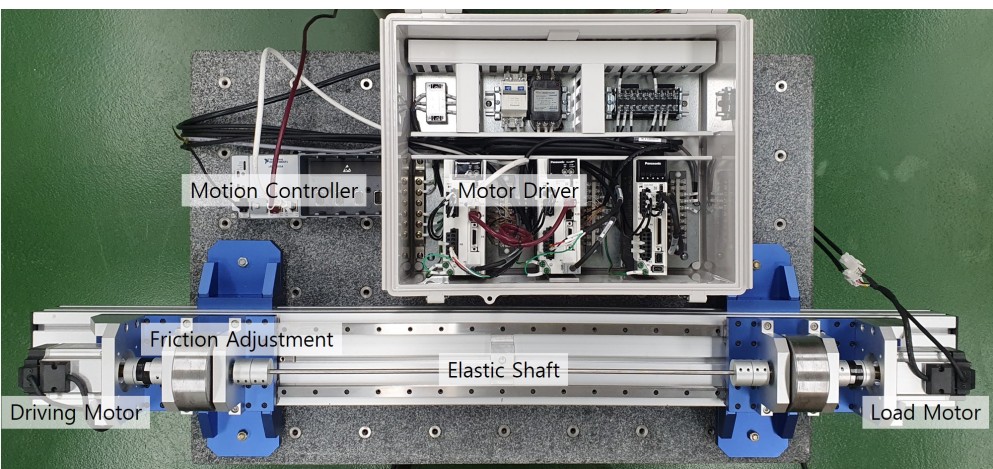

**Figure 3.** Experiment Setup.

The parameters of the two-inertia system, including the nonlinear stiffness and friction, are listed in Table 1. In addition, the link acceleration $\dot{\omega}_l$ of (6) is obtained from the encoder value of the load motor in Figure 3. In other words, the angle acceleration is obtained by differentiating the position value of the encoder twice. Figure 4 shows the block diagram of the overall system with the proposed state observer. Here, the block $\Omega(\hat{x})$ means the nonlinear terms of the friction and stiffness in the elastic joint model. The difference in design approach between the proposed method and the linear observer is the addition of the block $\Omega(\hat{x})$ and the design method of the observer gain $L$ that depends on the Lipschitz constant $\gamma$ of $\Omega(x)$, as shown in the Equations (7) and (9). The design parameters, including the observer gain $L$ of (8), are chosen as depicted in Table 2.

**Table 1.** Parameters of the experimental system.

| Parameter | Value | Unit |
|---|---|---|
| damping ($D$) | 600 | $\mathrm{Nm \cdot s/rad}$ |
| motor inertia ($J_m$) | 0.001027 | $\mathrm{kg \cdot m^2}$ |
| linear stiffness coefficient ($k_1$) | $1.5 \times 10^6$ | $\mathrm{Nm/rad}$ |
| nonlinear stiffness coefficient ($k_2$) | $9.85 \times 10^{11}$ | $\mathrm{Nm/rad^3}$ |
| breakpoint deflection ($\theta_B$) | 2 | arcmin |
| viscous friction coefficient ($f_v$) | 0.006 | $\mathrm{Nm \cdot s/rad}$ |
| coulomb friction coefficient ($f_c$) | 1.5 | Nm |
| $\mu_k$ | 0.5 | |
| $\beta$ | 0.5 | |
| $\alpha$ | 5 | |

**Table 2.** Design parameters for experiments.

| Parameter | Value |
|---|---|
| $\epsilon$ | $1.0 \times 10^{-6}$ |
| observer gain $L$ | $[3.2477, \ 3.0286, \ 1.6021, \ 6.6788]^T \cdot 10^2$ |

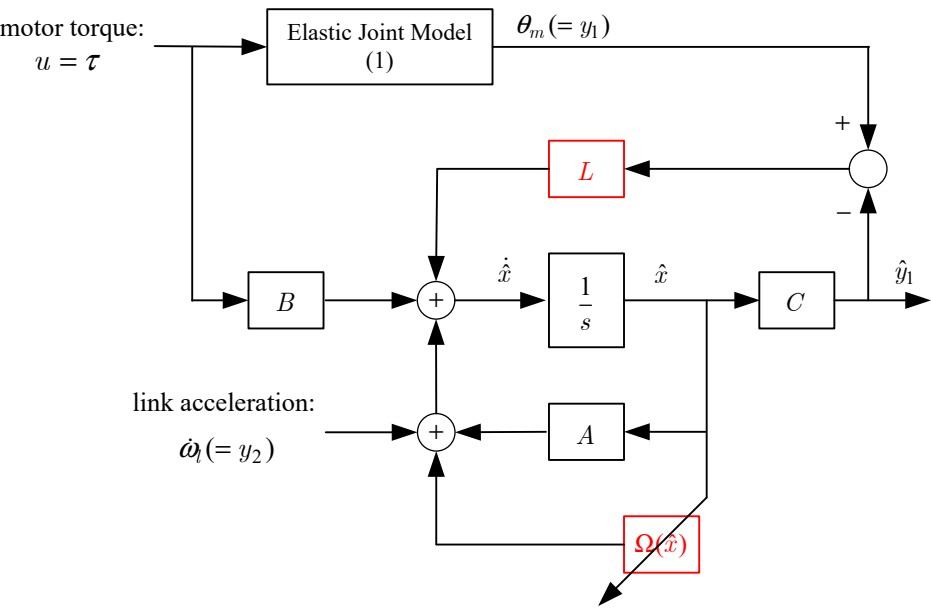

**Figure 4.** Block Diagram of the Overall System.

Now, as shown in Figure 5, we perform an experiment to compare the proposed observer considering the nonlinear stiffness and friction with the conventional observer considering only the nonlinear stiffness of [9]. The actual link trajectory (black solid line) controlled by the pre-installed PID controller is the measured value from the optical incremental encoder on the load motor. From the conventional and proposed observer, the estimated values of the actual link trajectory are indicated by the dash–dot (red color) line and dashed (blue color) line, respectively. For a more detailed performance comparison, we enlarge Figure 5 at 1.2 s and 6.2 s, as depicted in Figure 6. In order to evaluate the estimation performance, we calculate the RMSE (Root Mean Square Error) of the link

position estimation error for the conventional method and proposed approach, respectively, as follows.

$$\mathrm{RMSE}(\hat{\theta}_l, \theta_l) = \begin{cases} 124.8400 & \text{for conventional method,} \\ 7.8889 & \text{for proposed method.} \end{cases}$$

The comparison of the observation performance can also be confirmed from the estimation error of the actual link trajectory, as shown in Figure 7.

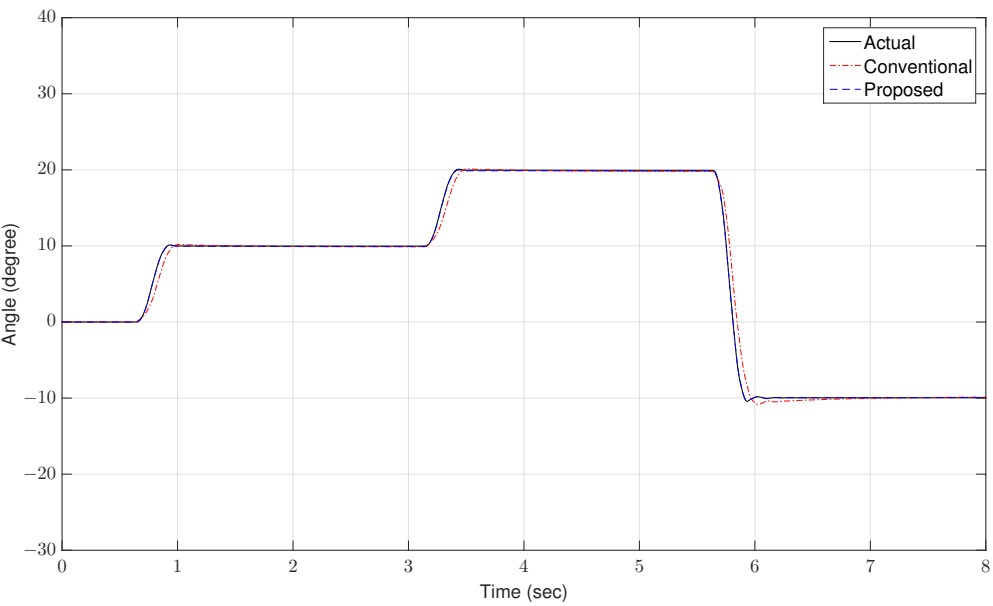

**Figure 5.** Observer performance comparison (link position).

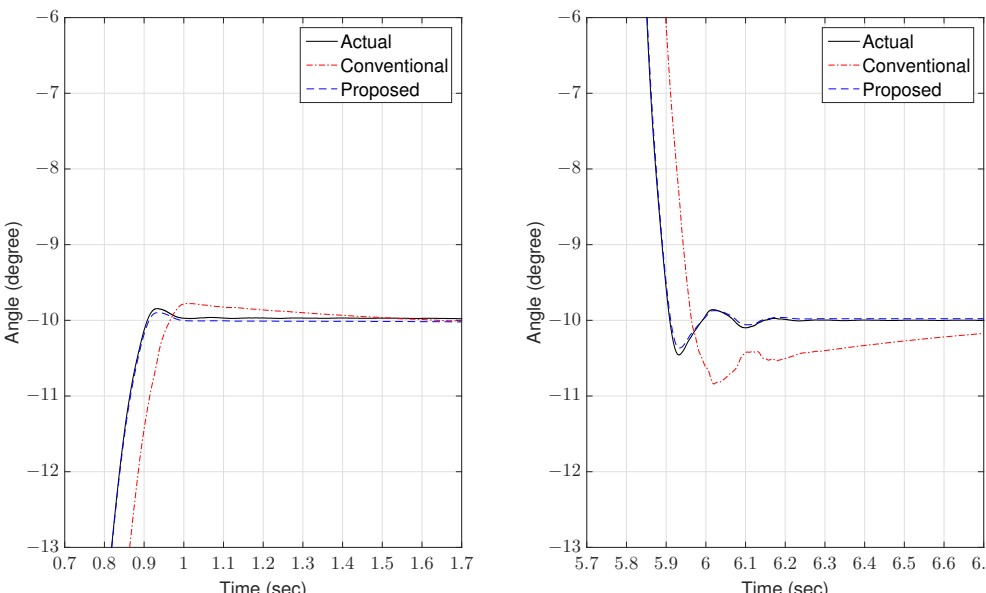

**Figure 6.** Enlarged version of Figure 5.

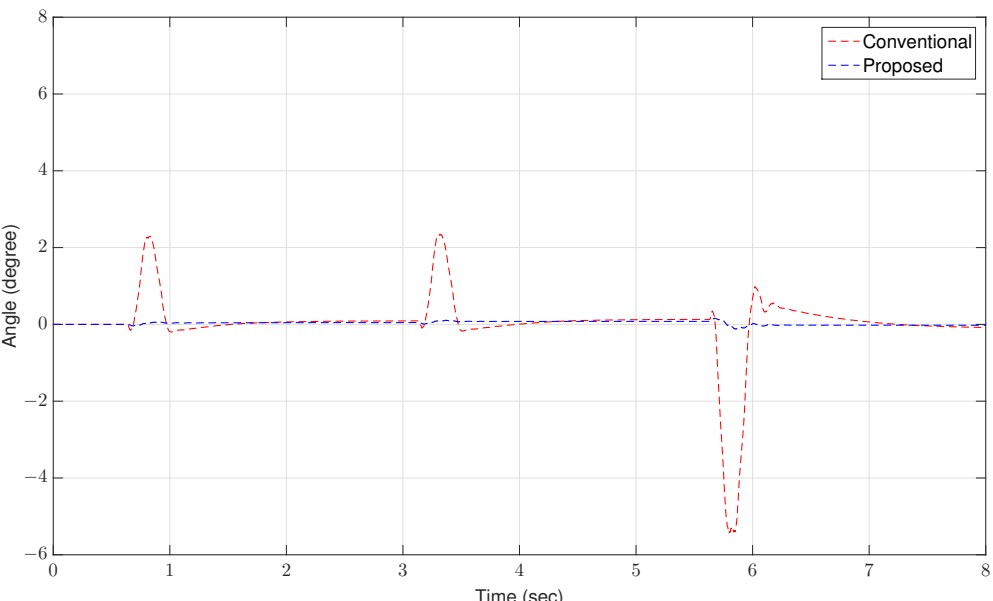

**Figure 7.** Estimation error of the link position.

## 5. Conclusions

In this brief, we have proposed a state estimator for an elastic joint with nonlinear friction based on LMI (linear matrix inequality) optimization. From the assumption that the elastic joint has a nonlinear friction model that is considered a smooth function, the proposed estimator has guaranteed that the estimation error of link position and velocity converges to zero. Although the positive definite solution satisfying the LMI cannot be found if the Lipschitz constant is too large, we have applied to a two-inertia system to validate the proposed method, and thus its performance has been verified. Finally, the nonlinear friction model could demonstrate jumping resonance phenomena that can appear in practical nonlinear systems [20]. Thus, further investigation seems to be needed in the future.

**Author Contributions:** K.-H.L. and H.K. proposed the concept idea, wrote the manuscript, and performed the experiments. T.-Y.K. edited manuscript. All authors have read and agreed to the published version of the manuscript.

**Funding:** This work was supported by the Institute of Information and Communications Technology's Planning and Evaluation (IITP) grant funded by the Korean government (MSIT) (No. 2022-0-00059, Development of an automated parcel unloading system using a mobile manipulator and AI-based object recognition technology).

**Institutional Review Board Statement:** Not applicable.

**Informed Consent Statement:** Not applicable.

**Data Availability Statement:** Not applicable.

**Conflicts of Interest:** The authors declare no conflict of interest.

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
