# Peer review of "State Observer Based on an Accelerometer for an Elastic Joint with Nonlinear Friction"

_applsci, doi:10.3390/app122412991_

Round 1
Reviewer 1 Report
The manuscript is well presented. I recommend it for publication.
Reviewer 2 Report
Please see the attachment

Reviewer 3 Report
This paper proposed a state estimator for an elastic joint with nonlinear friction based on LMI (linear matrix inequality) optimization. The proposed state observer guaranteed that the estimation error for position and velocity of the link connected via an elastic joint containing a nonlinear stiffness converges to zero. Finally, the authors tested the proposed method by experiments on a two-inertia system with an elastic shaft. The following comments need to be considered in the revision:
1. Some background should be enriched, such as the researchers on state observer for elastic joints.
2. The difference of the methods between this paper and the published work, for example, the literature [9], should be added in the revised paper.
3. The contribution of this paper is suggested to be given in the introduction.
4. There are some minors in References format. For example, 7 is not standard.
5. The writing should be double checked throughout the paper. For example, it should be “it” but not “It” in the line of 55 on page 3.
Reviewer 4 Report
This paper presents a state observer for an elastic joint with nonlinear friction via the information of acceleration sensor. Follows are some suggestions when the authors decided to revise the paper.
1) The motivation on the study has to be further stressed. The problems discussed in this paper have been extensively studied by others. Hence the importance of the particular problems in this paper and/or why revisit the problem should be clearly addressed.
2) Why no measurement noise or disturbances are considered for system (6)?
3) The results proposed in Theorem 1 are obtained by using the common Lyapunov function approach. In my opinion, the authors should point out the innovation.
4) More remarks should be added such that the readers can be well guided
5) The defects of the proposed method may be included in the Conclusion part.
6) The authors should do however a careful reading to correct some remaining minor typos and English wording issues
Round 2
Reviewer 2 Report
The reviewer appreciates the author's effort to improve the paper in the revised version. However, I have the following comment:
In the manuscript, the effect of noises is only bounded not minimized, while many existing approaches for the Lipschitz system can deal with the noise via H-infinity, H2, or mixed H2/H-infinity.... criterion. The proposed method should be compared with the existing approach to highlight the contribution.
